# Therapeutic Utility and Adverse Effects of Biologic Disease-Modifying Anti-Rheumatic Drugs in Inflammatory Arthritis

**DOI:** 10.3390/ijms232213913

**Published:** 2022-11-11

**Authors:** Hong Ki Min, Se Hee Kim, Hae-Rim Kim, Sang-Heon Lee

**Affiliations:** 1Division of Rheumatology, Department of Internal Medicine, Konkuk University Medical Center, Seoul 05030, Korea; 2Division of Rheumatology, Department of Internal Medicine, Research Institute of Medical Science, Konkuk University School of Medicine, Seoul 05030, Korea

**Keywords:** monoclonal antibody, biologic disease-modifying anti-rheumatic drug, rheumatoid arthritis, spondyloarthritis, psoriatic arthritis, inflammatory arthritis

## Abstract

Targeting specific pathologic pro-inflammatory cytokines or related molecules leads to excellent therapeutic effects in inflammatory arthritis, including rheumatoid arthritis, ankylosing spondylitis, and psoriatic arthritis. Most of these agents, known as biologic disease-modifying anti-rheumatic drugs (bDMARDs), are produced in live cell lines and are usually monoclonal antibodies. Several types of monoclonal antibodies target different pro-inflammatory cytokines, such as tumor necrosis factor-α, interleukin (IL)-17A, IL-6, and IL-23/12. Some bDMARDs, such as rituximab and abatacept, target specific cell-surface molecules to control the inflammatory response. The therapeutic effects of these bDMARDs differ in different forms of inflammatory arthritis and are associated with different adverse events. In this article, we summarize the therapeutic utility and adverse effects of bDMARDs and suggest future research directions for developing bDMARDs.

## 1. Introduction

Rheumatoid arthritis (RA), spondyloarthritis (SpA), and crystal-induced arthritis have a common pathogenesis, as they are provoked by pathologic inflammation-related innate and adaptive immune cells. An abnormally increased immune response and inflammation induce a local inflammatory response in affected joints and consequently promote systemic complications, such as cardiovascular diseases or disease-specific extra-articular manifestations, such as interstitial lung disease (ILD) in RA and uveitis/psoriasis/inflammatory bowel disease (IBD) in SpA. Levels of several pro-inflammatory cytokines are elevated in inflammatory arthritis, and the concept of blocking specific cytokines or cell-surface markers has led to the development of biologic disease-modifying anti-rheumatic drugs (bDMARDs). These bDMARDs have shown symptomatic improvement in patients with inflammatory arthritis and extra-articular symptoms.

In the present review, we aim to describe the therapeutic effects of bDMARDs in inflammatory arthritis as well as the adverse effects associated with each agent. Furthermore, we discuss the future directions of bDMARD development in the field of inflammatory arthritis.

## 2. Pathogenesis of Inflammatory Arthritis

The main immunological process and pathological site of inflammation differ in each type of inflammatory arthritis. The synovium is the main site where inflammatory cells and immunological processes occur in RA [1,2]. In gout, synovitis is the main pathological process when acute gouty arthritis occurs [3], and chronically elevated serum urate levels cause monosodium urate (MSU) deposits, called tophi [4]. However, SpA, which includes axial spondyloarthritis (axSpA), psoriatic arthritis (PsA), reactive arthritis, and IBD-related arthritis, shows inflammation at the enthesis where ligaments or tendons attach [5]. As the synovium is the site of major pathologic processes in RA and gout, bone erosion is the main consequence of RA [6] and gout [7]. In axSpA patients, abnormal new bone formation at the vertebral corner, syndesmophyte, is the main structural change, and this process can eventually cause total ankylosis of the spine [8]. Enthesitis of the anterior and posterior longitudinal ligaments of the spine is the main pathological process that causes syndesmophyte formation in patients with ankylosing spondylitis (AS), a prototype of axSpA [9]. Patients with PsA usually show a peripheral arthritis-dominant clinical pattern rather than axial joint inflammation [10]. In PsA, the underlying mechanism for nail psoriasis and dactylitis can also be explained by enthesitis of the nail bed and collateral ligaments of the digit [11]. 

The main pathological processes of each type of inflammatory arthritis differ. The adaptive immune cells including helper T cells, B cells, and plasma cells are assumed to be the main pathologic immune cells that induce RA, and therefore, RA is defined as an “autoimmune disease” [12,13]. In the era of type 1 helper T cells (Th1) and Th2, Th1 was thought to be the major disease inducer of RA [14]; however, after the discovery of Th17 and regulatory T cells (Tregs), the paradigm of RA pathogenesis changed toward an imbalanced Th17 and Treg population as the main pathologic process of RA [15,16,17]. Although the paradigm shift from “Th1/Th2 imbalance” to “Th17/Treg imbalance” has been introduced, IFN-γ-producing Th1 cells are still suggested as one of the RA-inducing immune cells, especially in the early stage of RA [18,19]. In addition, promotion of the Th2 response suppresses the inflammatory response in an RA mouse model [20]. On the other hand, innate immune cells also aid in RA pathogenesis. Macrophages and dendritic cells act as antigen-presenting cells, and present auto-reactive antigens, such as citrullinated peptides, to naïve CD4+ T cells to become pathologic helper T cell subsets [21]. The IL-23/IL-17 axis is the main pathological process in SpA [22]. The importance of the innate immune response is emphasized in SpA pathogenesis, and SpA is called an “autoinflammatory disease” rather than an “autoimmune disease” [23]. Activated antigen-presenting cells produce IL-23, which activates various immune cells, including type 3 innate lymphoid cells, invariant natural killer T cells, γδ T cells, RORγt+ CD3+ CD4− CD8− tissue-resident T cells, mucosa-associated invariant T cells, neutrophils, mast cells, and Th17 [22,24,25,26,27]. Although the major pathologic cells of SpA are still debated, actual effector cytokines that promote articular and extra-articular symptoms of SpA are interleukin (IL)-17, tumor necrosis factor (TNF)-α, and IL-22 [22]. Gout is an inflammatory arthritis induced by MSU crystals [28]. Acute gout flare is induced by NLRP3 inflammasome activation and the consequent conversion of pro-IL-1β and pro-IL-18 into the active forms of IL-1β and IL-18 [28]. Although major pathologic cells are still not definitively identified in each inflammatory arthritis or may differ between each inflammatory arthritis, they share common pro-inflammatory cytokines (IL-1β, IL-6, IL-17, TNF-α, et al.) as the main effector in their pathogenesis.

## 3. Clinical Effects of bDMARDs in RA, axSpA, PsA, and Gout

Currently, four different types of bDMARDs—TNF inhibitors, anti-IL-6R inhibitors, T cell costimulatory inhibitor (CTLA-Ig), and anti-CD20 Ab—have been approved for RA therapy [29,30,31]. After the development of several bDMARDs, the use of TNF inhibitors as a first-line bDMARD in RA has been reduced. [32] In axSpA, TNF inhibitors and anti-IL-17A Abs are used clinically as bDMARDs [31,33,34]. More bDMARDs are recommended for PsA treatment: (1) TNF inhibitors, (2) anti-IL-17A Abs, (3) IL-23/IL-12 (p40) inhibitor and IL-23 (p19) inhibitor, and (4) CTLA-Ig. In the management of gout, anti-IL-1 therapeutics, such as anakinra, rilonacept, and canakinumab, are conditionally recommended to resolve gout flare [35,36]. The classification and indications for each bDMARD are presented in Table 1.

### 3.1. Therapeutic Effects and Adverse Events of TNF Inhibitors

TNF inhibitors exert anti-arthritic effects by blocking the activity of TNF-α. Five TNF inhibitors have been approved for RA, axSpA, and PsA, with different molecular structures [37]. The most outstanding difference between the TNFR2-Fc fusion protein (etanercept) and monoclonal antibody forms (adalimumab, infliximab, golimumab, and certolizumab pegol) is that monoclonal antibody forms can bind to the monomer and trimer forms of TNF-α with higher affinity, whereas the TNFR2-Fc fusion protein only binds to the trimer form of TNF-α with relatively unstable binding affinity [38,39]. Another important difference is that the TNFR2-Fc fusion protein can block TNF-β (also called lymphotoxin-α) [38]. These differences are assumed to be the cause of the lack of therapeutic or preventive effects of the TNFR2-Fc fusion protein (etanercept) on uveitis and IBD of axSpA. Clinical trials of etanercept in Crohn’s disease revealed that etanercept was ineffective in Crohn’s disease [40]. In addition, secukinumab, an anti-IL-17A Ab, did not show clinical efficacy in clinical trials on Crohn’s disease [41]. Consequently, a monoclonal antibody form of TNF inhibitor is recommended in axSpA patients with IBD rather than TNFR2-Fc protein or anti-IL-17A Ab [33,34]. Regarding the aspect of acute anterior uveitis (AAU) of axSpA, the hazard ratio (HR) for AAU recurrence was significantly higher in the etanercept group than in the monoclonal antibody form of TNF inhibitors group [42,43]. Furthermore, the monoclonal antibody form of TNF inhibitors showed a lower risk of AAU than the anti-IL17A Ab in a meta-analysis [44]. The aforementioned findings suggest that treatment guidelines for axSpA recommend a monoclonal antibody form of TNF inhibitor over TNFR2-Fc fusion protein or anti-IL-17A Ab in patients with axSpA with a history of AAU [33,34].

In addition to the therapeutic effects of bDMARDs on inflammatory arthritis, the prevention of joint destruction in RA/PsA (assessed by total Sharp score) or syndesmophyte formation (assessed by modified stoke ankylosing spondylitis spinal score (mSASSS)) in axSpA is also important because these structural damages are important factors for QoL in patients with arthritis [45,46]. The use of TNF inhibitors has been shown to suppress spinal structural damage in patients with axSpA via direct [47] and indirect mechanisms [48]. Long-term follow-up data from AS patients showed that the period during TNF inhibitor usage showed lower mSASSS progression than in the period in which patients did not receive a TNF inhibitor, which proved the direct effect of TNF inhibitors on spinal structural damage in patients with AS [47]. Data from the Swiss Clinical Quality Management cohort revealed that patients with AS who were TNF inhibitor users, and who achieved clinical remission (ankylosing spondylitis disease activity score (ASDAS) < 1.3) showed less progression of mSASSS than TNF inhibitor users with higher ASDAS scores (ASDAS > 1.3). This implied that clinical remission using TNF inhibitors can predict the prevention of spinal structural damage [48]. In patients with RA, the use of TNF inhibitors suppressed radiographic progression of the hands and feet [49,50,51,52,53,54,55,56,57,58,59,60]. In addition, TNF inhibitors slowed radiographic progression in patients with PsA [61,62]. 

The leading cause of death in inflammatory arthritis is cardiovascular events, and the risk of cardiovascular events is higher in patients with RA, axSpA, PsA, and gout [63,64,65,66,67] than in the general population. Chronically increased levels of inflammatory mediators promote plaque formation and the progression of atherosclerosis [68]. Several observational studies have shown that patients taking TNF inhibitors have a lower risk of cardiovascular events than patients using only conventional synthetic DMARDs (csDMARDs) in RA, AS, and PsA [69,70,71]. However, in a specific subgroup of patients, it was shown that the preventive effect of TNF inhibitors on cardiovascular events may be lower than that of other bDMARDs. Insurance claim data from the USA showed that RA patients with a previous history of cardiovascular disease [72], comorbidity of type 2 diabetes mellitus [73], or age over 65 years [74] had a lower HR for cardiovascular events when using abatacept than when using TNF inhibitors. However, these were observational studies, and further studies are needed to clarify the differences between bDMARDs in the prevention of cardiovascular events. Three clinical trials (ATTACH, RECOVER, and RENAISSANCE) have been performed to test whether TNF inhibitors prevented heart failure (HF) aggravation [75,76]; these trials were conducted with the hypothesis that TNF-α is a pathologic cytokine contributing to HF progression [77]. However, using etanercept or infliximab in HF patients did not prevent HF aggravation, and even high doses of infliximab showed a worse prognosis for HF [75,76]. Therefore, the American College of Rheumatology (ACR) guidelines prefer non-TNF inhibitor bDMARDs to TNF inhibitors in RA patients with NYHA III to IV HF [29]. The additional therapeutic roles of bDMARDs including TNF inhibitors are summarized in Table 2.

The most common serious adverse event associated with TNF inhibitors is an infection. However, the risk of infection is already increased by inflammatory arthritis itself [95], concomitant use of glucocorticoids [95], and conventional synthetic DMARDs (csDMARDs) [96]. A meta-analysis that included 71 clinical trials showed 40% of serious infections were attributed to TNF inhibitor use [97]. However, a recent study showed that the hazard ratio for serious infection in patients with RA using methotrexate + TNF inhibitors was non-significant (HR = 1.23, 95% confidence interval [CI] 0.87–1.74) when compared with patients with RA using triple csDMARDs (methotrexate + sulfasalazine + hydroxychloroquine) [98], which supports that adding TNF inhibitors for patients with RA does not further increase infection risk. However, in the case of preoperative management, withdrawal of TNF inhibitors 1 week before surgery plus regular intervals for each TNF inhibitor (for example, 2 weeks for adalimumab) is recommended when total hip or knee arthroplasty is planned in TNF inhibitor users [99]. Another specific issue related to infectious diseases is tuberculosis, and many developing countries still have a high prevalence of tuberculosis [100]. The risk of tuberculosis was significantly increased by TNF inhibitors (incidence rate ratio = 3.61 with a 95% CI 1.38–8.07 for patients with RA and incidence rate ratio = 4.87 with a 95% CI 1.50–15.39 for patients with AS) [101]. In addition, a meta-analysis also showed a significantly increased risk for tuberculosis occurrence (odds ratio = 1.94, 95% CI 1.10–3.44) in TNF inhibitor users [102]. Screening tests for latent tuberculosis by chest radiographic imaging and interferon gamma release assays for tuberculosis/PPD skin tests are mandatory before initiating TNF inhibitors in intermediate to high-burden areas for tuberculosis [31]. The underlying mechanism was suggested to be that the TNF inhibitor (infliximab) depletes specific immune cells (CD45RA^+^ effector memory CD8^+^ T cells), which are critical for the clearance of tuberculosis-infected macrophages [103]. TNF receptor knockout mice showed an increased burden of tuberculosis reactivation and granuloma formation compared with wild-type mice [104]. These results suggested that TNF-α plays a critical role in tuberculosis clearance. The risk of tuberculosis was lower in etanercept users than in users of monoclonal antibody forms of TNF inhibitors in Asia [105], which may arise from the relatively low affinity of etanercept for TNF-α compared with the monoclonal antibody forms of TNF inhibitors [38].

TNF was discovered by Lloyd J. Old, who demonstrated that cytokines derived from macrophages had cytotoxic effects on mouse fibrosarcoma cells [106]. Thereafter, various functions of TNF-α were revealed, which showed pathological roles in inflammatory arthritis. Paradoxically, in spite of the term “TNF”, TNF inhibitor use did not increase the occurrence of cancer in patients with RA (incidence rate ratio = 0.913, *p* = 0.546) [107], or increase the risk of cancer recurrence (HR = 1.06, 95% CI 0.73–1.54 for pooled cancer data, HR = 1.1 with a 95% CI 0.4–2.8 for breast cancer) [108,109]. However, in non-melanoma skin cancer, a meta-analysis including 10 prospective observational studies demonstrated that TNF inhibitor use in patients with RA increased the risk of non-melanoma skin cancer (pooled relative risk = 1.28, 95% CI 1.19–1.38) [110]. Therefore, TNF inhibitors are relatively safe with respect to cancer risk; however, in the era of high skin cancer prevalence [111], we should be aware of non-melanoma skin cancer when using TNF inhibitors. 

TNF inhibitors are also currently used in psoriasis and are effective in the skin manifestation of PsA [112]. The increased level of TNF-α, which is produced by conventional dendritic cells and Th17 in the skin, is thought to be the key pathogenesis of classical skin psoriasis [113], and this is the mechanism by which TNF inhibitors reduce skin psoriasis. However, using TNF inhibitors increases the risk of paradoxical psoriasis occurrence by approximately 1.915 times that of non-TNF inhibitor users [114]. The underlying pathology of “paradoxical psoriasis” is assumed to involve T cell-independent and plasmacytoid-dendritic-cell-driven IFN-α-dependent mechanisms [115].

Although the incidence of ILD in patients with RA is low (2.7–3.8 cases per 100,00 patients), patients with RA with ILD have a higher disease burden and poor prognosis [116,117]. The mortality related to respiratory complications and cancer was significantly higher in patients with RA with ILD than in patients with RA without ILD (HR = 4.39, 95% CI 4.13–4.67 for respiratory complications, HR = 1.56, 95% CI 1.43–1.71 for cancer) [117]. Some case reports have shown ILD occurrence or ILD aggravation after TNF inhibitor treatment initiation in patients with RA. However, insurance claim data including 11,219 patients with RA showed that the HR for ILD was similar for TNF inhibitors and for other bDMARDs (rituximab, abatacept, and tocilizumab) [118]. In another cohort study, the odds ratio (OR) for ILD in the TNF inhibitor group was insignificant when compared with that in csDMARD users (OR = 1.03, 95% CI 0.51–2.07) [119]. The adverse events of TNF inhibitors and other bDMARDs are presented in Table 3.

### 3.2. Therapeutic Use and Adverse Events of Anti-IL-6R Abs

Two IL-6 receptor blocking agents, tocilizumab and sarilumab, have proven therapeutic effects in RA [122,133],. The drug failure of tocilizumab was lower than the TNF inhibitor in RA patients [134]. Tocilizumab and sarilumab decreased C-reactive protein (CRP) levels but did not achieve a sufficient clinical response in patients with AS [135,136]. Anti-IL-6R Abs have not been tested in clinical trials for PsA, but in a case series of PsA patients in whom TNF inhibitors failed, tocilizumab was insufficient for PsA symptoms [137]. Therefore, anti-IL-6R Abs are only recommended for RA treatment [29] but not for axSpA or PsA.

Using anti-IL-6R Ab (tocilizumab or sarilumab) with methotrexate showed less radiographic progression than methotrexate alone [78,79,80,81]. Combination therapy with tocilizumab and methotrexate was superior in preventing radiographic progression of RA compared with tocilizumab monotherapy [82]. Tocilizumab monotherapy showed comparable, but not superior, therapeutic efficacy with methotrexate monotherapy [138], and combination therapy with tocilizumab and methotrexate showed better outcomes on radiographic progression than monotherapy [82]. Therefore, clinical guidelines recommend the use of anti-IL-6R Abs with methotrexate in patients with RA [29,31].

The preventive role of tocilizumab on cardiovascular events was similar to etanercept (HR = 1.05, 95% CI 0.77–1.43) [83]. Another study showed that tocilizumab was comparable with adalimumab and etanercept in terms of cardiovascular event prevention, but tocilizumab had lower risk when compared with infliximab (HR = 1.61, 95% CI 1.22–2.12, reference group tocilizumab user) [84]. Furthermore, the HR of tocilizumab for acute myocardial infarction (AMI) and major adverse cardiovascular events (MACE) was significantly lower than for rituximab in RA patients who failed TNF inhibitors (HR = 0.12, 95% CI 0.02–0.56 for AMI, HR = 0.41, 95% CI 0.23–0.72 for MACE) [85], and the HR was lower for coronary heart disease than abatacept in RA patients aged over 65 years (HR = 0.64, 95% CI 0.41–0.99) [74]. Additionally, in the Corrona RA registry, older age was associated with tocilizumab monotherapy [139].

The IL-6 signaling pathway is one of the main stimulants for CRP production [140], and, interestingly, anti-IL-6R Abs have shown a greater reduction in serum CRP levels than other bDMARDs. In the MONARCH trial, sarilumab showed greater CRP reduction than adalimumab (94% vs. 24%) [141]. Another study showed that the tocilizumab group had lower levels of CRP and disease activity score–28 joints (DAS28)-CRP than the adalimumab group; however, the ultrasound-based synovitis score was similar in the two groups [142]. Although anti-IL-6R Abs failed in clinical trials of AS, the serum CRP levels were significantly decreased [135,136]. Therefore, CRP reduction and disease activity scores based on CRP may overestimate the therapeutic effects of anti-IL-6R Abs. However, in another study, the magnitude of CRP reduction at 24 weeks after initiation of tocilizumab was positively correlated with achieving remission [143].

The most common serious adverse event of anti-IL-6R Abs is infection [123,144]. The risk of serious infection was comparable between tocilizumab users and TNF inhibitor users (HR = 1.05, 95% CI 0.95–1.16) [120]. However, the risk of serious bacterial infection, skin and soft tissue infection, and diverticulitis was significantly higher in the tocilizumab group than in the TNF inhibitor group [120,121]. In addition, a lower risk of urological and gynecological infections was found in the tocilizumab group than in the TNF inhibitor group [121]. One study showed that none of the patients with RA treated with tocilizumab experienced tuberculosis reactivation [145]. This implies that the risk of tuberculosis reactivation is rare with tocilizumab and that it is relatively safer than TNF inhibitors in terms of tuberculosis.

Changes in blood cell count and liver enzymes are other common adverse events of anti-IL-6R Abs [123]. Grade 1/2 or 3/4 neutropenia occurred in 28.2 and 3.1% of the tocilizumab group, which was higher than the placebo group (8.9% and 0.2%, respectively); however, the incidence rate of neutropenia-related infection was not higher in patients with RA taking tocilizumab [124]. Neutropenia occurred within 6 weeks after tocilizumab treatment and did not proceed thereafter [124]. One study showed 12.3% (14/114) of patients with RA taking tocilizumab had thrombocytopenia, and most of them had grade 1 thrombocytopenia [125]. The liver enzyme (aspartate transaminase (AST) and alanine transaminase (ALT)) levels were elevated in most of the tocilizumab-treated groups; however, elevation over 3-fold the upper limit of the reference range was only observed in 2% of the tocilizumab group [122]. Comparison between tocilizumab monotherapy and methotrexate monotherapy showed that ALT/AST elevations were similar in the two groups, and also that severe elevation (more than 3-fold higher than the reference range) of AST/ALT was only seen in approximately 1% of the tocilizumab group [126]. These laboratory changes (neutropenia, thrombocytopenia, and liver enzyme elevation) were reversible after halting tocilizumab treatment [122,124].

### 3.3. Therapeutic Use and Adverse Events of T Cell Costimulatory Inhibitor (CTLA-Ig, Abatacept)

The T cell costimulatory inhibitor abatacept is a fusion protein of CTLA-4 and the Fc portion of human IgG1 [146]. Abatacept showed therapeutic efficacy in patients with RA and PsA [88,147,148] but not in patients with AS [149]. AS and PsA belong to the SpA category and share various susceptibility genes; however, the clinical efficacy of abatacept differs. In addition, abatacept decreased arthritis symptoms assessed by the ACR 20 response but did not improve psoriasis in patients with PsA [88].

Joint destruction in RA was suppressed by abatacept treatment. The abatacept + methotrexate group showed less radiographic progression than the methotrexate monotherapy group (mean change in total Sharp score 0.63 vs. 1.06, *p* = 0.040) [86]. Radiographic progression was more pronounced when abatacept treatment was maintained for a longer duration (total Sharp score change for years 2 to 3 vs. years 1 to 2, 0.25 vs. 0.43, *p* = 0.022) [87]. The proportion of radiographic non-progression was also higher in PsA patients from the abatacept group than in the placebo group (42.7 vs. 32.7%, *p* = 0.034) [88].

With regard to cardiovascular disease (CVD), abatacept showed a lower risk of stroke, HF, and MACE than rituximab in patients with RA who had previously failed TNF inhibitors [85]. In bDMARD-naïve RA patients, abatacept showed a lower risk of cardiovascular events than TNF inhibitors in specific subgroups with a history of CVD, type 2 diabetes mellitus, and in elderly patients (age ≥ 65 years) [72,73,74]. One observational study demonstrated that the risk of cardiovascular events between abatacept and tocilizumab was comparable [84], whereas in elderly RA patients, aged over 65 years, tocilizumab showed less risk for cardiovascular events than abatacept (HR = 0.65, 95% CI 0.41–0.99) [74].

Serious infection is the most common serious adverse event in abatacept users, and pooled data from five clinical trials of RA showed that pneumonia, urinary tract infection, and gastroenteritis were the leading opportunistic infections after abatacept treatment [127]. However, a meta-analysis revealed that the pooled ORs for serious infections in abatacept users were non-significant when compared with csDMARD users (OR = 1.35, 95% CI 0.78–2.32) [128], and the risk for hospitalized infection was lower than TNF inhibitor users in patients with RA (HR = 0.78, 95% CI 0.64–0.95) [129]. Furthermore, two cohort studies showed that none of the abatacept users showed tuberculosis reactivation [150,151]. Therefore, in terms of opportunistic infections, abatacept is comparable with csDMARDs and safer than TNF inhibitors.

### 3.4. Therapeutic Application and Adverse Events of Anti-IL-17A/IL-17R Abs

Two anti-IL-17A Abs (secukinumab and ixekizumab) have been approved for use in patients with axSpA and PsA [33,112]. Recently, brodalumab, an anti-IL-17 receptor A blocking Ab, showed significant clinical improvement in patients with axSpA and PsA in a phase 3 clinical trial [152,153]. Serum IL-17 and Th17 levels were increased in patients with RA [154] and suppressed in response to TNF inhibitors [155]. Therefore, anti-IL-17A Ab (secukinumab) was administered to patients with RA, and a high dose of secukinumab (150 mg every 4 weeks) showed a higher achievement rate for the primary endpoint (ACR20 response) than the placebo group [156]. However, a higher dose of secukinumab was inferior to abatacept (ACR20 response 30.7 vs. 42.8%, respectively), and the secondary outcome (ACR50 response) was comparable with the placebo group [156]. Another anti-IL-17A Ab, ixekizumab, showed better clinical improvement than the placebo group in a phase II study [157], but a phase III study has not yet been conducted. None of the anti-IL-17A/IL-17R Abs have been approved for RA treatment.

The 2-year follow-up data for secukinumab in PsA patients showed 81 to 89% of secukinumab users had no radiographic progression, and the mean change in total Sharp score was lower with a high dose of secukinumab than a low dose [91]. In the 3-year follow-up data from the SPIRIT-P1 study, 71% of patients taking ixekizumab every 4 weeks and 61% of patients taking ixekizumab every 2 weeks were non-progressors [92]. In patients with AS, 72.5 and 82.1% of secukinumab users showed mSASSS changes of less than two units [89]. In patients with axSpA, 89.6% of ixekizumab users were non-progressors over 2 years [90]. However, the effect of anti-IL17A Ab on radiographic changes in axSpA or PsA patients was not compared with proper control groups, and more cumulative data may be needed to clarify the preventive role of anti-IL-17A Ab on radiographic progression in patients with axSpA or PsA.

With regard to CVD, only short-term data on anti-IL-17A Abs are available [158], and it is impossible to conclude the effect of anti-IL-17A Ab on CVD of inflammatory arthritis.

Infection is the most notable adverse event associated with anti-IL-17A antibodies. IL-17 is essential for the defense mechanism of the host against pathogens and plays an important role in fungal infections [159]. Pooled data from seven clinical studies, including patients with axSpA, PsA, and psoriasis, showed that the incidence rates of serious infection and *Candida* infection were 1.2 to 1.9 and 0.7 to 2.2 per 100 patient-years [130]. The risk of *Candida* infection was 4- to 10-fold higher with anti-IL-17A Abs than with TNF inhibitors, and the requirement for antifungal agents increased 2- to 16-fold after anti-IL17A Ab was started [131]. However, tuberculosis activation was not reported in a pooled cohort study that included 12,319 patients with axSpA/PsA/psoriasis [160].

Two phase II studies of IL-17 blocking agents (secukinumab and brodalumab) have been conducted on Crohn’s disease, and they were terminated early due to aggravation of disease activity or a higher incidence rate of serious adverse events [41,161]. Although two pooled datasets of secukinumab showed that the incidence rate of new-onset IBD was very low (0.01–0.05 per 100 patient-years and 0.05–0.4 per 100 patient-years) [130,162], paradoxical new onset of IBD was reported in the secukinumab and ixekizumab treatment arms of AS, PsA, and psoriasis patients [163]. Therefore, the ACR and EULAR guidelines still recommend monoclonal TNF inhibitors over anti-IL-17A Ab in SpA patients with IBD features [33,34]. 

### 3.5. Therapeutic Application and Adverse Events of Anti-IL-23/IL-12 p40 and Anti-IL-23 p19 Abs

Anti-IL-23/IL-12 p40 Ab (ustekinumab) and anti-IL-23 p19 Ab (guselkumab and risankizumab) have been approved for PsA treatment [164,165,166,167,168]. However, ustekinumab and risankizumab treatments did not meet the primary and secondary endpoints for patients with axSpA [169,170]. Several hypotheses for axSpA failure have been proposed as follows: (1) IL-23 is not a critical cytokine in the clinical phase of axSpA, (2) Th17 may not be the main pathologic immune cell in axSpA, (3) IL-17 production in axSpA may be independent of IL-23, and 4) IL-23 has no role in enthesitis, which is the main pathologic site of axSpA [171]. Similar to axSpA, IL-23 blocking agents have failed to treat RA [172].

A recent meta-analysis of 48 studies showed that serious adverse events associated with ustekinumab were rare [173]. Nasopharyngitis, headache, and upper respiratory tract infection were the most common adverse events in patients with PsA taking ustekinumab in phase III clinical studies [164,165]. Guselkumab and risankizumab (anti-IL-23 p19 Abs) also show serious adverse events, and serious infections are rare in patients with PsA [166,167,168]. Other bDMARDs had a higher risk of hospitalization due to serious infection when compared with ustekinumab (HR = 1.66 [95% CI 1.34–2.06] for adalimumab, HR = 1.39 [95% CI 1.01–1.90] for etanercept, HR = 1.74 [95% CI 1.00–3.03] for golimumab, HR = 2.92 [95% CI 1.80–4.72] for infliximab, HR = 2.98 [95% CI 1.20–7.41] for ixekizumab, and HR = 1.84 [95% CI 1.24–2.72] for secukinumab) [93].

In insurance claim data from French patients, the risk of acute coronary syndrome and stroke were significantly increased in ustekinumab users with high cardiovascular risk (OR = 4.17, 95% CI 1.19–14.59), whereas it was non-significant in the low cardiovascular risk group (OR = 0.30, 95% CI 0.03–3.13) [132]. 

### 3.6. Therapeutic Application and Adverse Events of Anti-CD20 Ab

In contrast to other bDMARDs, rituximab is injected as a cycle (on days 1 and 15) as an indication for RA [174]. One course of rituximab treatment with methotrexate showed significant clinical improvement compared with placebo + methotrexate [174,175]. The mean change in the total Sharp score over 2 years was lower in the rituximab + methotrexate group than in the placebo + methotrexate group (1.14 vs. 2.81, *p* < 0.0001) [94]. The preventive effect of rituximab on CVD was comparable with that of csDMARDs [69]. In RA patients who failed TNF inhibitors, the HR for cardiovascular events was lower in the tocilizumab and abatacept groups than in the rituximab group [85]. 

The adverse events risk in the rituximab group was comparable with the placebo group in a meta-analysis (pooled relative risk = 1.062, 95% CI 0.912–1.236) [176]. In addition, pooled data showed that ORs for infection and serious infection were insignificant when comparing rituximab and non-rituximab groups of RA patients [177].

### 3.7. Therapeutic Application and Adverse Events of Anti-IL-1 Blocker

Anakinra, an IL-1R receptor antagonist, was approved for use in RA patients [178]. A meta-analysis revealed that ACR20 response achievement was higher in the anakinra group than the placebo group (relative risk = 1.42, 95% CI 1.01–2.00) [179]. However, treatment-related discontinuation was higher in the anakinra group than in the placebo group [179]. Furthermore, anti-IL-1 blockers are not commonly used in practice, and an anti-IL-1 blocker has been excluded from recent RA treatment guidelines [29,30]. IL-1β plays a crucial role in acute gout flare [28] and anti-IL-1 blockers have been used for gout treatment. Anakinra showed meaningful pain reduction in patients with acute gout flares, but the therapeutic effects were not superior to those of glucocorticoids [180]. Combination therapy with rilonacept and a non-steroidal anti-inflammatory drug (NSAID) did not provide additional pain relief compared with NSAID monotherapy in acute gout flare [181]. Rilonacept reduced approximately 70% more gout flare events than placebo after initiation of urate-lowering therapy [182], but this study did not compare its preventive role with a proper positive control such as an NSAID, glucocorticoids, or colchicine. Pooled data from two clinical trials showed the superiority of canakinumab for pain relief within 72 h and prevention of new flares compared with a glucocorticoid, but serious adverse events were significantly higher in the canakinumab group than the glucocorticoid group (8.0 vs. 3.5%) [183]. These agents have not been approved by the US FDA because the harm caused by anti-IL-1 blockers (increased risk of infection) may not exceed the benefit. In addition, ACR guidelines only conditionally recommend anti-IL-1 blockers when other oral agents (NSAID, glucocorticoids, and colchicine) for gout flares are intolerable or contraindicated [35]. 

## 4. Future Treatment Modalities

### 4.1. Dual Cytokine Blockage

Recently, methods for maximizing the treatment efficacy and reducing adverse events of bDMARDs have been attempted. First, blocking of multiple pro-inflammatory cytokines in inflammatory arthritis has been attempted in RA and PsA. Bispecific blocking antibodies against TNF-α and IL-17 reduced the inflammatory response more effectively than a single blocking agent for each cytokine in RA-FLS [184]. The dual cytokine inhibitor ABT-122, which blocks TNF-α and IL-17A, was tested in patients with RA and PsA [185,186]. Although the clinical efficacy was higher in the ABT-122 group than in the placebo group, it was not superior to monotherapy with the TNF inhibitor adalimumab [185,186]. In an animal model of SpA, HLA-B27/human β2-microglobulin transgenic rats, dual blockage of TNF-α and IL-17A reduced arthritis severity and bone loss, but the anti-arthritic effects were not superior to a single blockage against TNF-α or IL-17A [187]. Dual blocking antibodies for IL-17A and IL-6R have been developed [188]; however, this has not been attempted in inflammatory arthritis. IL-17 can be subdivided into six subtypes (IL-17A–F), among which IL-17A and IL-17-F are the most potent pro-inflammatory cytokines [189]. Bimekizumab, a dual blocking antibody against IL-17A and IL-17F, showed clinical improvement in patients with AS and PsA in phase 2 clinical studies [190,191]. In patients with AS, bimekizumab showed a superior ASAS40 response to that in the placebo group in a phase 2 study [190]. In a phase 2 study of PsA patients, bimekizumab achieved an ACR 50 response in 57.7% of PsA patients [191], but results from a phase 3 study are not yet published. In most cases of inflammatory arthritis, various pro-inflammatory cytokines are upregulated, and the network of these cytokines promotes the progression of inflammatory arthritis. Therefore, inhibition of multiple pathologic cytokines may be more useful for controlling the pathogenesis and progression of inflammatory arthritis; however, the dosage and combination to block cytokines should be further examined. 

### 4.2. Nanoparticle-Based Cytokine Inhibitors

Another novel treatment strategy for inflammatory arthritis is to minimize the adverse effects of bDMARDs. Currently available bDMARDs affect the whole body due to systematic administration. Targeting only specific sites such as the synovium or enthesis can reduce systemic adverse events with maximal therapeutic effects. In addition, target-organ-specific bDMARDs may reduce the risk of developing anti-drug antibodies, which reduce the clinical efficacy of bDMARDs [192]. Currently, nanoparticle-based drug systems are used to deliver specific drugs to targeted tissues in cancer or autoimmune diseases [193]. Drug delivery systems based on nanoparticles not only increase precise targeting by enhancing the permeability and retention of drugs on targeted organs but also improve the stability and biocompatibility of the drug [193]. One study developed albumin-based nanomedicine by using the high affinity of albumin and secreted protein acidic and rich in cysteine (SPARC), which is abundantly expressed in RA synovium [194]. This albumin–methotrexate fusion nanoparticle showed increased retention in the paw of an RA mouse model compared with systemic administration of methotrexate [194]. Fusion proteins of hyaluronate–gold nanoparticles and tocilizumab reduced arthritis severity and pro-inflammatory cytokine production in a CIA mouse model [195]. Although many preclinical trials of nanoparticle-based medication have been attempted [196,197], none of them have been approved for use in patients with inflammatory arthritis. Further studies of nanoparticle-based drug delivery systems may provide both better therapeutic efficacy and reduced adverse events in these patients.

## 5. Conclusions

Currently, various bDMARDs are used to treat inflammatory arthritis and they have improved the prognosis and QoL of these patients. Inflammatory arthritis is a systemic disease, and various extra-articular symptoms and systemic complications can occur. Therefore, the additional effects of bDMARDs on extra-articular symptoms or systemic complications should be considered when choosing appropriate bDMARDs in individual patients. Furthermore, the development of novel bDMARDs for dual cytokine inhibition and the use of target-organ-specific delivery systems may enhance clinical efficacy and reduce the frequency and severity of adverse events.

## Figures and Tables

**Table 1 ijms-23-13913-t001:** Classification and indications for bDMARDs.

	bDMARDs	Indication
TNF inhibitors	Etanercept Adalimumab Infliximab Golimumab Certolizumab pegol	RA AS PsA
Anti-IL-6R Abs	Tocilizumab Sarilumab	RA
T cell costimulatory inhibitors	Abatacept	RA PsA
Anti-IL-17A Abs	Secukinumab Ixekizumab	AS PsA
Anti-IL-17R Abs	Brodalumab	AS PsA
Anti-IL-23/IL-12 p40 Abs	Ustekinumab	PsA
Anti-IL-23 p19 Abs	Guselkumab Risankizumab	PsA
Anti-CD20 Ab	Rituximab	RA
Anti-IL-1 blockers	Anakinra Rilonacept Canakinumab	RA (only in anakinra) Gout

**Table 2 ijms-23-13913-t002:** Additional effects of bDMARDs in patients with inflammatory arthritis over anti-arthritic effect.

bDMARD	Inflammatory Arthritis	Additional Effect	Reference
TNF inhibitor	RA	Prevention of joint destruction	[49,50,51,52,53,54,55,56,57,58,59,60]
	RA	Cardiovascular event preventive effect superior to csDMARDs	[69,70,71]
	RA	Cardiovascular event preventive effect inferior to T cell costimulatory inhibitor (abatacept) in specific subgroup (previous history of cardiovascular disease, type 2 DM, and old age)	[72,73,74]
	AS	Suppression of spinal structural progression	[47,48]
	AS	Prevention of AAU in monoclonal Ab form of TNF inhibitor	[42,43,44]
	AS	Cardiovascular event preventive effect	[71]
	PsA	Prevention of joint destruction	[61,62]
	PsA	Cardiovascular event preventive effect	[71]
Anti-IL-6R Abs	RA	Prevention of joint destruction is superior in anti-IL-6R Ab + methotrexate compared with monotherapy of methotrexate or anti-IL-6R Ab	[78,79,80,81,82]
	RA	Cardiovascular event preventive effect comparable with or superior to TNF inhibitor	[83,84,85]
	RA	Cardiovascular event preventive effect superior to anti-CD20 Ab or T cell costimulatory inhibitors	[74,85]
T cell costimulatory inhibitor	RA	Prevention of joint destruction	[86,87]
	RA	Cardiovascular event preventive effect superior to anti-CD20 Ab (TNF inhibitor non-responder) and superior to TNF inhibitor (previous history of cardiovascular disease, type 2 DM, and old age)	[72,73,74,85]
	PsA	Prevention of joint destruction	[88]
Anti-IL-17A/IL-17R Abs	AS	Suppression of spinal structural progression	[89,90]
	PsA	Prevention of joint destruction	[91,92]
Anti-IL-23/IL-12 p40 and anti-IL-23 p19 Abs	PsA	Lower risk for hospitalization due to serious infection than TNF inhibitors or anti-IL-17A Abs	[93]
Anti-CD20 Ab	RA	Prevention of joint destruction	[94]

**Table 3 ijms-23-13913-t003:** Specific adverse events of bDMARDs in patients with inflammatory arthritis.

bDMARD	Inflammatory Arthritis	Adverse Events	Reference
TNF inhibitor	RA	Increased infection risk (esp. tuberculosis)	[97,101]
	RA	Tuberculosis risk: lower in etanercept than other TNF inhibitors (in Asia)	[105]
	RA	Increased risk of non-melanoma skin cancer	[110]
	RA	Cardiovascular event preventive effect inferior to T cell costimulatory inhibitor (abatacept) in specific subgroup (previous history of CVD, T2DM, and old age)	[72,73,74]
	AS	Increased infection risk of tuberculosis	[101]
	AS	Tuberculosis risk: lower in etanercept than other TNF inhibitors (in Asia)	[105]
	AS	Paradoxical psoriasis	[114]
Anti-IL-6R Abs	RA	General infection risk: comparable with TNF inhibitors Increased risk of serious bacterial infection, skin/soft tissue infection, and diverticulitis compared with TNF inhibitors	[120,121]
	RA	Non-severe and reversible neutropenia, thrombocytopenia, and AST/ALT elevation	[122,123,124,125,126]
T cell costimulatory inhibitor	RA	Increased infection risk (especially pneumonia, urinary tract infection, and gastroenteritis)	[127]
	RA	Infection risk comparable with csDMARDs and lower than TNF inhibitor	[128,129]
Anti-IL-17A/IL-17R Abs	AS/PsA	Increased infection risk (especially *Candida* infection)	[130,131]
Anti-IL-23/IL-12 p40 and anti-IL-23 p19 Abs	PsA	Increased risk of cardiovascular event in patients with baseline high cardiovascular risk	[132]

## Data Availability

Not applicable.

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
