# Peer review of "Therapeutic Utility and Adverse Effects of Biologic Disease-Modifying Anti-Rheumatic Drugs in Inflammatory Arthritis"

_ijms, 2022, doi:10.3390/ijms232213913_

Round 1
Reviewer 1 Report
Dear authors,
The article is very well written. Thus, I must aware you some recent high impact references that you should consider to add to enrich your review.
https://www.nature.com/articles/s41598-021-94504-x Interestingly depicts about the recent decrease in the choice of TNF for RA.
https://www.sciencedirect.com/science/article/pii/S0049017221001128 Compares failures between 1st and 2nd line of bDMARD.
https://arthritis-research.biomedcentral.com/articles/10.1186/s13075-020-02408-4 published by a solid research group in the RA field, supports the TCZ use which is strongly discussed in your work.
https://doi.org/10.1039/D2NA00229A This very recent review about new advances in nanoparticle approaches for drug delivery-based systems for RA treatment could increase your final remarks in lines 497-498.
After such minor adjustments, I must congratulate to authors for the careful attention in writing and accurate appointments. Best regards
Author Response
The article is very well written. Thus, I must aware you some recent high impact references that you should consider to add to enrich your review.
https://www.nature.com/articles/s41598-021-94504-x Interestingly depicts about the recent decrease in the choice of TNF for RA.
https://www.sciencedirect.com/science/article/pii/S0049017221001128 Compares failures between 1st and 2nd line of bDMARD.
https://arthritis-research.biomedcentral.com/articles/10.1186/s13075-020-02408-4 published by a solid research group in the RA field, supports the TCZ use which is strongly discussed in your work.
https://doi.org/10.1039/D2NA00229A This very recent review about new advances in nanoparticle approaches for drug delivery-based systems for RA treatment could increase your final remarks in lines 497-498.
After such minor adjustments, I must congratulate to authors for the careful attention in writing and accurate appointments. Best regards
Answer : Thanks for recommending proper reference, and we added recommended articles in revised manuscript (track change, reference number 32, 105, 118, 197).
Reviewer 2 Report
Dear Authors,
Your manuscript presents a study interesting and significant for its clinical aspects. For patients with moderate or severe rheumatoid arthritis for whom methotrexate therapy has failed or who are intolerant to methotrexate, there is uncertainty about which replacement therapy to try - conventional synthetic disease-modifying anti-rheumatic drugs (alone or in combination), biologic drugs (including biosimilars) and synthetic targeted disease-modifying anti-rheumatic drugs appear to be effective for different desired effects. A comparative review of the clinical effectiveness and safety of biologic drugs (including biosimilars), targeted DMARDs, and combination therapies with conventional DMARDs will help inform treatment choice.
My recommendation is to accept manuscript for publication after minor revision (typographical errors in References section). For better understanding, I have included my comments in the text of the manuscript, and I have listed some of them below:
1/ The review could not determine if one treatment provided a greater benefit than the other because there was no data on all treatments for each outcome, and often there were few differences when comparing the outcomes of these treatments directly.
2/ It should be emphasized that the results of the review are limited to the short term, no evidence from observational studies was incorporated into the review, and the majority of included studies were at high or unclear risk of bias. The results should therefore be interpreted with caution.
3/ References:
Authors should carefully prepare the References section, as there are many inaccuracies in it.
The formatting of the references is heterogeneous. I recommend preparing the references with a bibliography software package in line with Int. J. Mol. Sci. requirements. Although there are no strict formatting requirements, the minor shortcomings in the formatting of the literature items exist:
- Some journal names are given in full, including many not written correctly - each element of the journal title should be written with a capital letter; some journal names are correctly abbreviated. I have included some comments (not all!) in the text of the manuscript.
- The Authors should add more recent references to their list (if it's possible). Almost 47% of items are older than 5 years.

Author Response
My recommendation is to accept manuscript for publication after minor revision (typographical errors in References section). For better understanding, I have included my comments in the text of the manuscript, and I have listed some of them below:
Answer: We revised the reference section by using Endnote style of MDPI (https://endnote.com/style_download/mdpi/).
1/ The review could not determine if one treatment provided a greater benefit than the other because there was no data on all treatments for each outcome, and often there were few differences when comparing the outcomes of these treatments directly.
Answer: We agree with the referee’s comment, and revised some sentence which mentioned the difference of bDMARDs too definitely (track change, line 159, 163-165).
2/ It should be emphasized that the results of the review are limited to the short term, no evidence from observational studies was incorporated into the review, and the majority of included studies were at high or unclear risk of bias. The results should therefore be interpreted with caution.
Answer: We agree with the referee’s comment, and revised some sentence which mentioned the difference of bDMARDs too definitely (track change, line 159, 163-165).
3/ References:
Authors should carefully prepare the References section, as there are many inaccuracies in it.
The formatting of the references is heterogeneous. I recommend preparing the references with a bibliography software package in line with Int. J. Mol. Sci. requirements. Although there are no strict formatting requirements, the minor shortcomings in the formatting of the literature items exist:
- Some journal names are given in full, including many not written correctly - each element of the journal title should be written with a capital letter; some journal names are correctly abbreviated. I have included some comments (not all!) in the text of the manuscript.
- The Authors should add more recent references to their list (if it's possible). Almost 47% of items are older than 5 years.
Answer: Answer: We revised the reference section by using Endnote style of MDPI (https://endnote.com/style_download/mdpi/). In addition, we tried to use more recent reference in revised manuscript (added reference number 32, 105, 118, 197), and erased some old references.
Reviewer 3 Report
A great deal of work has been done, but the article is presented in a perplexed way. Many facts are presented in this article, but in a way in which they do not represent novelty, but only facts, facts and more facts, that have not been analyzed. By citing so many details from different studies, are the authors sure that they are comparable?
No classification of the drug groups and their representatives is presented. This could have been done in Table 1.
There is a lot of unnecessary repetition (e.g. Two IL-6 receptor-blocking agents, tocilizumab and sarilumab, have proven therapeutic effects in RA [103,104], and these agents suppress the IL-6 mediated inflammatory response by binding to the IL-6 receptor) and I get the impression that the authors are not very clear what exactly they want to express with these facts from the presented literature.
The authors are explaining things that have already been elucidated and are already enshrined in the recommendations of clinical guidelines. I get the impression that they are explaining to themselves how scientists have already come to their assumed conclusions. At certain points, their explanations sound explanatory (e.g. 260 - 'CRP, which is produced in the liver, is the most widely used acute-phase reactant in real world clinical settings').
The entire paragraph from 260 to 273 makes no sense (CRP reduction may predict the clinical response to tocilizumab in patients with RA.) 418-419 "RA is a prototype of autoimmune systemic arthritis that typically presents with autoantibodies [13]."
2. Pathogenesis of inflammatory arthritis - presents a series of facts but provides no new information.
The sentence (82-84) „Although major pathologic cells are still not definitively identified in each inflammatory arthritis or may differ between each inflammatory arthritis, they share common proinflammatory cytokines as the main effector in their pathogenesis" - provides no information because the authors have not bothered to summarize which these "common proinflammatory cytokines" are.
The text in section 3.2 is largely redundant since it is presented in Table 2.
Т. 3.4 from 275 to 286 - the authors contradict themselves
Speaking of adverse effects, the authors should also consider the published data in the current SPC of mentioned medications.
Paragraphs 421 to 429 do not provide any new information - "In contrast to other bDMARDs, rituximab is injected as a cycle (on days 1 and 15) as an indication of RA [177]. One course of rituximab treatment with methotrexate showed significant clinical improvement compared with placebo + methotrexate [177,178]. The mean change in the total Sharp score over 2 years was lower in the rituximab + methotrexate group than in the placebo + methotrexate group (1.14 vs 2.81, p < 0.0001) [179]. The preventive effect of rituximab on CVD was comparable to that of csDMARDs [69]. In RA patients who failed TNF inhibitors, the HR for cardiovascular events was lower in the tocilizumab and abatacept groups than in the rituximab group [116]."
Just as 3.11 and 3.12 are combined, so could be 3.9 and 3.10, 3.7 and 3.8, 3.5 and 3.6, 3.3 and 3.4, 3.1 and 3.2.
The authors give no explanation of this statement 440-441 - "Two other anti-IL-1 blockers, rilonacept and canakinumab, have been developed; however, none have been approved for RA, PsA, or axSpA.", and there is no citation. Why they have not been approved for RA, PsA, or axSpA? Why do the authors not discuss the fact? New drugs are registered on the basis of data from clinical trials. It was enough for the authors to mention registered indications, by making a well-ordered classification table.
In 4. Future treatment modalities - The section is very poorly presented.
„Bimekizumab, a dual blocking antibody against IL-17A and IL-17F, showed clinical improvement in patients with AS and PsA in phase 2 clinical trials [194,195].“ Two large-scale studies can hardly be summarized in one sentence. The authors have not given what the indications of bimekizumab are.
In conclusion, the article presents a lot of details related to the mechanism of action, however the information regarding indications and side effects is incomplete. The review represents a well-known information and there are other excellent review articles already published on the theme. The authors need to systematize and analyse the presented facts. The conclusion is uncertain. The review does not help to resolve the treatment issues due to heterogeneity and complex pathology of mentioned diseases.
Author Response
A great deal of work has been done, but the article is presented in a perplexed way. Many facts are presented in this article, but in a way in which they do not represent novelty, but only facts, facts and more facts, that have not been analyzed. By citing so many details from different studies, are the authors sure that they are comparable?
Answer: The anti-inflammatory and anti-arthritis effects of bDMARDs are already reviewed in many review articles. However, the treatment target of inflammatory arthritis not only focus on controlling arthritis, but also aim to reduce extra-articular symptoms / prognosis. We tried to focus on the impact of bDMARDs on extra-articular or real-world evidence based safety data of each bDMARDs. Controlled clinical trials are needed to evaluate direct comparison of each bDMARDs on long term prognosis, but these are almost impossible. And only observational studies are available when comparing effects on long-term prognosis or extra-articular symptoms.
No classification of the drug groups and their representatives is presented. This could have been done in Table 1.
Answer: We newly made Table 1 as grouping bDMARDs in revised manuscript (track change, line 98).
There is a lot of unnecessary repetition (e.g. Two IL-6 receptor-blocking agents, tocilizumab and sarilumab, have proven therapeutic effects in RA [103,104], and these agents suppress the IL-6 mediated inflammatory response by binding to the IL-6 receptor) and I get the impression that the authors are not very clear what exactly they want to express with these facts from the presented literature.
Answer: We erased repetition in revised manuscript (Track change 104-112, 152-156, 314-320, 399-401, 426-431).
The authors are explaining things that have already been elucidated and are already enshrined in the recommendations of clinical guidelines. I get the impression that they are explaining to themselves how scientists have already come to their assumed conclusions. At certain points, their explanations sound explanatory (e.g. 260 - 'CRP, which is produced in the liver, is the most widely used acute-phase reactant in real world clinical settings').
Answer: We erased the sections where it is not necessary (Track change 104-112, 152-156, 314-320, 399-401, 426-431).
The entire paragraph from 260 to 273 makes no sense (CRP reduction may predict the clinical response to tocilizumab in patients with RA.) 418-419 "RA is a prototype of autoimmune systemic arthritis that typically presents with autoantibodies [13]."
Answer: We erased the sentence (track change, line 282-283, 426-431)
- Pathogenesis of inflammatory arthritis - presents a series of facts but provides no new information.
Answer: This is a review article which should be based on already known facts. And brief explanation of known pathogenesis could be helpful to readers to understand the article.
The sentence (82-84) „Although major pathologic cells are still not definitively identified in each inflammatory arthritis or may differ between each inflammatory arthritis, they share common proinflammatory cytokines as the main effector in their pathogenesis" - provides no information because the authors have not bothered to summarize which these "common proinflammatory cytokines" are.
Answer: We added common proinflammatory cytokines in revised manuscript (track change, line 84).
The text in section 3.2 is largely redundant since it is presented in Table 2.
Answer: The Table 2 is summary of the adverse events which is dominantly presented in each bDMARDs.
Т. 3.4 from 275 to 286 - the authors contradict themselves
Answer: The infection risk is commonly increased in bDMARDs user, and indicating specific infection risk in specific bDMARDs is relevant information.
Speaking of adverse effects, the authors should also consider the published data in the current SPC of mentioned medications.
Answer: Thanks for recommendation. We already saw SPC of each bDMARDs when writing the manuscript.
Paragraphs 421 to 429 do not provide any new information - "In contrast to other bDMARDs, rituximab is injected as a cycle (on days 1 and 15) as an indication of RA [177]. One course of rituximab treatment with methotrexate showed significant clinical improvement compared with placebo + methotrexate [177,178]. The mean change in the total Sharp score over 2 years was lower in the rituximab + methotrexate group than in the placebo + methotrexate group (1.14 vs 2.81, p < 0.0001) [179]. The preventive effect of rituximab on CVD was comparable to that of csDMARDs [69]. In RA patients who failed TNF inhibitors, the HR for cardiovascular events was lower in the tocilizumab and abatacept groups than in the rituximab group [116]."
Answer: The review article should be based on already published data.
Just as 3.11 and 3.12 are combined, so could be 3.9 and 3.10, 3.7 and 3.8, 3.5 and 3.6, 3.3 and 3.4, 3.1 and 3.2.
Answer: We combined the therapeutic application and adversed events of each bDMARDs in revised manuscrip (track change ).
The authors give no explanation of this statement 440-441 - "Two other anti-IL-1 blockers, rilonacept and canakinumab, have been developed; however, none have been approved for RA, PsA, or axSpA.", and there is no citation. Why they have not been approved for RA, PsA, or axSpA? Why do the authors not discuss the fact? New drugs are registered on the basis of data from clinical trials. It was enough for the authors to mention registered indications, by making a well-ordered classification table.
Answer: We added Table 1 as sorting bDMARDs and their indication (track change, line 98). And erased the aforementioned sentence (track change, lin 450-452).
In 4. Future treatment modalities - The section is very poorly presented.
„Bimekizumab, a dual blocking antibody against IL-17A and IL-17F, showed clinical improvement in patients with AS and PsA in phase 2 clinical trials [194,195].“ Two large-scale studies can hardly be summarized in one sentence. The authors have not given what the indications of bimekizumab are.
Answer: We added detailed data of both clinical studies in revised manuscript (Track change 485-488). And these were only phase 2 study, therefore, bimekizumab is not approved by US FDA or EMA, yet.
Round 2
Reviewer 3 Report
The authors have made some specific corrections to the text but have not revised the entire text. The material is so extensive that it is difficult to summarize so much data in one article. Due to this fact, the written review article is very confused. A lot of well-known facts are presented, but without summarizing or analyzing the data in any direction.
I do not agree that the information contained in the SPC of products should be presented in a review article (rituximab example).
Bimekizumab has been registered in the EU since August 2021, but the authors were unaware of this.
My opinion on the article remains that it contributes nothing.